# Wider and Deeper, Cheaper and Faster: Tensorized LSTMs for Sequence Learning

**Zhen He**[1,2], **Shaobing Gao**[3], **Liang Xiao**[2], **Daxue Liu**[2], **Hangen He**[2], and **David Barber**[1,4*]

[1]University College London, [2]National University of Defense Technology, [3]Sichuan University, [4]Alan Turing Institute

## Abstract

Long Short-Term Memory (LSTM) is a popular approach to boosting the ability of Recurrent Neural Networks to store longer term temporal information. The *capacity* of an LSTM network can be increased by widening and adding layers. However, usually the former introduces additional parameters, while the latter increases the runtime. As an alternative we propose the *Tensorized LSTM* in which the hidden states are represented by *tensors* and updated via a *cross-layer convolution*. By increasing the tensor size, the network can be widened efficiently without additional parameters since the parameters are shared across different locations in the tensor; by delaying the output, the network can be deepened implicitly with little additional runtime since deep computations for each timestep are merged into temporal computations of the sequence. Experiments conducted on five challenging sequence learning tasks show the potential of the proposed model.

## 1   Introduction

We consider the time-series prediction task of producing a desired output $\boldsymbol{y}_t$ at each timestep $t \in \{1, \ldots, T\}$ given an observed input sequence $\boldsymbol{x}_{1:t} = \{\boldsymbol{x}_1, \boldsymbol{x}_2, \cdots, \boldsymbol{x}_t\}$, where $\boldsymbol{x}_t \in \mathbb{R}^R$ and $\boldsymbol{y}_t \in \mathbb{R}^S$ are vectors[1]. The Recurrent Neural Network (RNN) [17, 43] is a powerful model that learns how to use a hidden state vector $\boldsymbol{h}_t \in \mathbb{R}^M$ to encapsulate the relevant features of the entire input history $\boldsymbol{x}_{1:t}$ up to timestep $t$. Let $\boldsymbol{h}_{t-1}^{cat} \in \mathbb{R}^{R+M}$ be the concatenation of the current input $\boldsymbol{x}_t$ and the previous hidden state $\boldsymbol{h}_{t-1}$:

$$\boldsymbol{h}_{t-1}^{cat} = [\boldsymbol{x}_t, \boldsymbol{h}_{t-1}] \tag{1}$$

The update of the hidden state $\boldsymbol{h}_t$ is defined as:

$$\boldsymbol{a}_t = \boldsymbol{h}_{t-1}^{cat} \boldsymbol{W}^h + \boldsymbol{b}^h \tag{2}$$

$$\boldsymbol{h}_t = \phi(\boldsymbol{a}_t) \tag{3}$$

where $\boldsymbol{W}^h \in \mathbb{R}^{(R+M) \times M}$ is the weight, $\boldsymbol{b}^h \in \mathbb{R}^M$ the bias, $\boldsymbol{a}_t \in \mathbb{R}^M$ the hidden activation, and $\phi(\cdot)$ the element-wise tanh function. Finally, the output $\boldsymbol{y}_t$ at timestep $t$ is generated by:

$$\boldsymbol{y}_t = \varphi(\boldsymbol{h}_t \boldsymbol{W}^y + \boldsymbol{b}^y) \tag{4}$$

where $\boldsymbol{W}^y \in \mathbb{R}^{M \times S}$ and $\boldsymbol{b}^y \in \mathbb{R}^S$, and $\varphi(\cdot)$ can be any differentiable function, depending on the task.

However, this vanilla RNN has difficulties in modeling long-range dependencies due to the vanishing/exploding gradient problem [4]. Long Short-Term Memories (LSTMs) [19, 24] alleviate

these problems by employing memory cells to preserve information for longer, and adopting gating mechanisms to modulate the information flow. Given the success of the LSTM in sequence modeling, it is natural to consider how to increase the complexity of the model and thereby increase the set of tasks for which the LSTM can be profitably applied.

We consider the *capacity* of a network to consist of two components: the *width* (the amount of information handled in parallel) and the *depth* (the number of computation steps) [5]. A naive way to widen the LSTM is to increase the number of units in a hidden layer; however, the parameter number scales quadratically with the number of units. To deepen the LSTM, the popular Stacked LSTM (*s*LSTM) stacks multiple LSTM layers [20]; however, runtime is proportional to the number of layers and information from the input is potentially lost (due to gradient vanishing/explosion) as it propagates vertically through the layers.

In this paper, we introduce a way to both widen and deepen the LSTM whilst keeping the parameter number and runtime largely unchanged. In summary, we make the following contributions:

(a) We tensorize RNN hidden state vectors into higher-dimensional tensors which allow more flexible parameter sharing and can be widened more efficiently without additional parameters.

(b) Based on (a), we merge RNN deep computations into its temporal computations so that the network can be deepened with little additional runtime, resulting in a *Tensorized RNN (tRNN)*.

(c) We extend the *t*RNN to an LSTM, namely the *Tensorized LSTM (tLSTM)*, which integrates a novel memory cell convolution to help to prevent the vanishing/exploding gradients.

## 2 Method

### 2.1 Tensorizing Hidden States

It can be seen from (2) that in an RNN, the parameter number scales quadratically with the size of the hidden state. A popular way to limit the parameter number when widening the network is to organize parameters as higher-dimensional tensors which can be factorized into lower-rank sub-tensors that contain significantly fewer elements [6, 15, 18, 26, 32, 39, 46, 47, 51], which is is known as tensor factorization. This implicitly widens the network since the hidden state vectors are in fact broadcast to interact with the tensorized parameters. Another common way to reduce the parameter number is to share a small set of parameters across different locations in the hidden state, similar to Convolutional Neural Networks (CNNs) [34, 35].

We adopt parameter sharing to cutdown the parameter number for RNNs, since compared with factorization, it has the following advantages: (i) *scalability*, i.e., the number of shared parameters can be set independent of the hidden state size, and (ii) *separability*, i.e., the information flow can be carefully managed by controlling the *receptive field*, allowing one to shift RNN deep computations to the temporal domain (see Sec. 2.2). We also explicitly tensorize the RNN hidden state vectors, since compared with vectors, tensors have a better: (i) *flexibility*, i.e., one can specify which dimensions to share parameters and then can just increase the size of those dimensions without introducing additional parameters, and (ii) *efficiency*, i.e., with higher-dimensional tensors, the network can be widened faster w.r.t. its depth when fixing the parameter number (see Sec. 2.3).

For ease of exposition, we first consider 2D tensors (matrices): we tensorize the hidden state $\boldsymbol{h}_t \in \mathbb{R}^M$ to become $\boldsymbol{H}_t \in \mathbb{R}^{P \times M}$, where $P$ is the *tensor size*, and $M$ the *channel size*. We locally-connect the first dimension of $\boldsymbol{H}_t$ in order to share parameters, and fully-connect the second dimension of $\boldsymbol{H}_t$ to allow global interactions. This is analogous to the CNN which fully-connects one dimension (e.g., the RGB channel for input images) to globally fuse different feature planes. Also, if one compares $\boldsymbol{H}_t$ to the hidden state of a Stacked RNN (*s*RNN) (see Fig. 1(a)), then $P$ is akin to the number of stacked hidden layers, and $M$ the size of each hidden layer. We start to describe our model based on 2D tensors, and finally show how to strengthen the model with higher-dimensional tensors.

### 2.2 Merging Deep Computations

Since an RNN is already *deep* in its temporal direction, we can deepen an input-to-output computation by associating the input $\boldsymbol{x}_t$ with a (delayed) future output. In doing this, we need to ensure that the output $\boldsymbol{y}_t$ is *separable*, i.e., not influenced by any future input $\boldsymbol{x}_{t'}$ ($t' > t$). Thus, we concatenate the projection of $\boldsymbol{x}_t$ to the *top* of the previous hidden state $\boldsymbol{H}_{t-1}$, then gradually shift the input

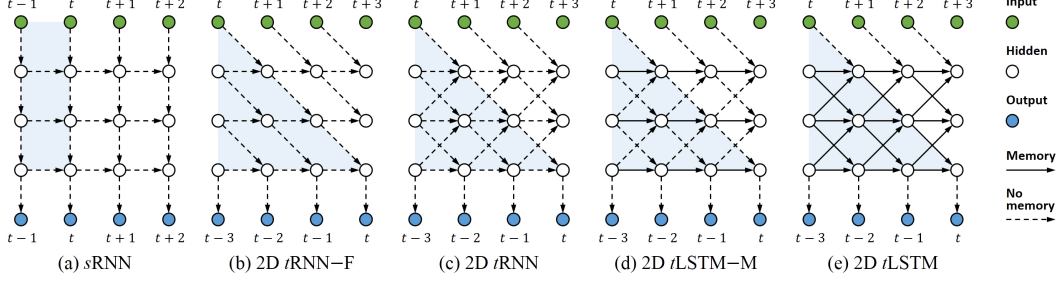

Figure 1: Examples of $s$RNN, $t$RNNs and $t$LSTMs. (a) A 3-layer $s$RNN. (b) A 2D $t$RNN without (−) feedback (F) connections, which can be thought as a *skewed* version of (a). (c) A 2D $t$RNN. (d) A 2D $t$LSTM without (−) memory (M) cell convolutions. (e) A 2D $t$LSTM. In each model, the blank circles in column 1 to 4 denote the hidden state at timestep $t−1$ to $t+2$, respectively, and the blue region denotes the receptive field of the current output $y_t$. In (b)-(e), the outputs are delayed by $L−1=2$ timesteps, where $L=3$ is the depth.

information down when the temporal computation proceeds, and finally generate $y_t$ from the *bottom* of $H_{t+L−1}$, where $L−1$ is the number of delayed timesteps for computations of *depth L*. An example with $L=3$ is shown in Fig. 1(b). This is in fact a *skewed sRNN* as used in [1] (also similar to [48]). However, our method does not need to change the network structure and also allows different kinds of interactions as long as the output is separable, e.g, one can increase the local connections and use feedback (see Fig. 1(c)), which can be beneficial for $s$RNNs [10]. In order to share parameters, we update $H_t$ using a convolution with a learnable kernel. In this manner we increase the complexity of the input-to-output mapping (by delaying outputs) and limit parameter growth (by sharing transition parameters using convolutions).

To describe the resulting $t$RNN model, let $H_{t−1}^{cat} \in \mathbb{R}^{(P+1)\times M}$ be the concatenated hidden state, and $p \in \mathbb{Z}_+$ the *location* at a tensor. The channel vector $h_{t−1,p}^{cat} \in \mathbb{R}^M$ at location $p$ of $H_{t−1}^{cat}$ is defined as:

$$h_{t−1,p}^{cat} = \begin{cases} x_t W^x + b^x & \text{if } p = 1 \\ h_{t−1,p−1} & \text{if } p > 1 \end{cases} \tag{5}$$

where $W^x \in \mathbb{R}^{R\times M}$ and $b^x \in \mathbb{R}^M$. Then, the update of tensor $H_t$ is implemented via a convolution:

$$A_t = H_{t−1}^{cat} \circledast \{W^h, b^h\} \tag{6}$$
$$H_t = \phi(A_t) \tag{7}$$

where $W^h \in \mathbb{R}^{K\times M^i\times M^o}$ is the *kernel weight* of size $K$, with $M^i = M$ input channels and $M^o = M$ output channels, $b^h \in \mathbb{R}^{M^o}$ is the *kernel bias*, $A_t \in \mathbb{R}^{P\times M^o}$ is the hidden activation, and $\circledast$ is the convolution operator (see Appendix A.1 for a more detailed definition). Since the kernel convolves across different hidden layers, we call it the *cross-layer convolution*. The kernel enables interaction, both bottom-up and top-down across layers. Finally, we generate $y_t$ from the channel vector $h_{t+L−1,P} \in \mathbb{R}^M$ which is located at the *bottom* of $H_{t+L−1}$:

$$y_t = \varphi(h_{t+L−1,P} W^y + b^y) \tag{8}$$

where $W^y \in \mathbb{R}^{M\times S}$ and $b^y \in \mathbb{R}^S$. To guarantee that the *receptive field* of $y_t$ only covers the current and previous inputs $x_{1:t}$ (see Fig. 1(c)), $L$, $P$, and $K$ should satisfy the constraint:

$$L = \left\lceil \frac{2P}{K − K \bmod 2} \right\rceil \tag{9}$$

where $\lceil \cdot \rceil$ is the ceil operation. For the derivation of (9), please see Appendix B.

We call the model defined in (5)-(8) the *Tensorized RNN (tRNN)*. The model can be widened by increasing the tensor size $P$, whilst the parameter number remains fixed (thanks to the convolution). Also, unlike the $s$RNN of runtime complexity $O(TL)$, $t$RNN breaks down the runtime complexity to $O(T+L)$, which means either increasing the sequence length $T$ or the network depth $L$ would not significantly increase the runtime.

## 2.3 Extending to LSTMs

To allow the $t$RNN to capture long-range temporal dependencies, one can straightforwardly extend it to an LSTM by replacing the $t$RNN tensor update equations of (6)-(7) as follows:

$$[\boldsymbol{A}_t^g, \boldsymbol{A}_t^i, \boldsymbol{A}_t^f, \boldsymbol{A}_t^o] = \boldsymbol{H}_{t-1}^{cat} \circledast \{\boldsymbol{W}^h, \boldsymbol{b}^h\} \tag{10}$$

$$[\boldsymbol{G}_t, \boldsymbol{I}_t, \boldsymbol{F}_t, \boldsymbol{O}_t] = [\phi(\boldsymbol{A}_t^g), \sigma(\boldsymbol{A}_t^i), \sigma(\boldsymbol{A}_t^f), \sigma(\boldsymbol{A}_t^o)] \tag{11}$$

$$\boldsymbol{C}_t = \boldsymbol{G}_t \odot \boldsymbol{I}_t + \boldsymbol{C}_{t-1} \odot \boldsymbol{F}_t \tag{12}$$

$$\boldsymbol{H}_t = \phi(\boldsymbol{C}_t) \odot \boldsymbol{O}_t \tag{13}$$

where the kernel $\{\boldsymbol{W}^h, \boldsymbol{b}^h\}$ is of size $K$, with $M^i = M$ input channels and $M^o = 4M$ output channels, $\boldsymbol{A}_t^g, \boldsymbol{A}_t^i, \boldsymbol{A}_t^f, \boldsymbol{A}_t^o \in \mathbb{R}^{P \times M}$ are activations for the new content $\boldsymbol{G}_t$, input gate $\boldsymbol{I}_t$, forget gate $\boldsymbol{F}_t$, and output gate $\boldsymbol{O}_t$, respectively, $\sigma(\cdot)$ is the element-wise sigmoid function, $\odot$ is the element-wise multiplication, and $\boldsymbol{C}_t \in \mathbb{R}^{P \times M}$ is the memory cell. However, since in (12) the previous memory cell $\boldsymbol{C}_{t-1}$ is only gated along the temporal direction (see Fig. 1(d)), long-range dependencies from the input to output might be lost when the tensor size $P$ becomes large.

**Memory Cell Convolution.** To capture long-range dependencies from multiple directions, we additionally introduce a novel *memory cell convolution*, by which the memory cells can have a larger receptive field (see Fig. 1(e)). We also dynamically generate this convolution kernel so that it is both time- and location-dependent, allowing for flexible control over long-range dependencies from different directions. This results in our $t$LSTM tensor update equations:

$$[\boldsymbol{A}_t^g, \boldsymbol{A}_t^i, \boldsymbol{A}_t^f, \boldsymbol{A}_t^o, \boldsymbol{A}_t^q] = \boldsymbol{H}_{t-1}^{cat} \circledast \{\boldsymbol{W}^h, \boldsymbol{b}^h\} \tag{14}$$

$$[\boldsymbol{G}_t, \boldsymbol{I}_t, \boldsymbol{F}_t, \boldsymbol{O}_t, \boldsymbol{Q}_t] = [\phi(\boldsymbol{A}_t^g), \sigma(\boldsymbol{A}_t^i), \sigma(\boldsymbol{A}_t^f), \sigma(\boldsymbol{A}_t^o), \varsigma(\boldsymbol{A}_t^q)] \tag{15}$$

$$\boldsymbol{W}_t^c(p) = \text{reshape}(\boldsymbol{q}_{t,p}, [K, 1, 1]) \tag{16}$$

$$\boldsymbol{C}_{t-1}^{conv} = \boldsymbol{C}_{t-1} \circledast \boldsymbol{W}_t^c(p) \tag{17}$$

$$\boldsymbol{C}_t = \boldsymbol{G}_t \odot \boldsymbol{I}_t + \boldsymbol{C}_{t-1}^{conv} \odot \boldsymbol{F}_t \tag{18}$$

$$\boldsymbol{H}_t = \phi(\boldsymbol{C}_t) \odot \boldsymbol{O}_t \tag{19}$$

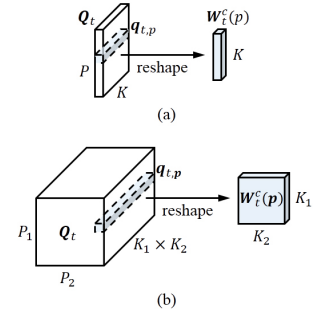

where, in contrast to (10)-(13), the kernel $\{\boldsymbol{W}^h, \boldsymbol{b}^h\}$ has additional $\langle K \rangle$ output channels[2] to generate the activation $\boldsymbol{A}_t^q \in \mathbb{R}^{P \times \langle K \rangle}$ for the dynamic kernel bank $\boldsymbol{Q}_t \in \mathbb{R}^{P \times \langle K \rangle}$, $\boldsymbol{q}_{t,p} \in \mathbb{R}^{\langle K \rangle}$ is the vectorized adaptive kernel at the location $p$ of $\boldsymbol{Q}_t$, and $\boldsymbol{W}_t^c(p) \in \mathbb{R}^{K \times 1 \times 1}$ is the dynamic kernel of size $K$ with a single input/output channel, which is reshaped from $\boldsymbol{q}_{t,p}$ (see Fig. 2(a) for an illustration). In (17), *each channel* of the previous memory cell $\boldsymbol{C}_{t-1}$ is convolved with $\boldsymbol{W}_t^c(p)$ whose values vary with $p$, forming a *memory cell convolution* (see Appendix A.2 for a more detailed definition), which produces a convolved memory cell $\boldsymbol{C}_{t-1}^{conv} \in \mathbb{R}^{P \times M}$. Note that in (15) we employ a softmax function $\varsigma(\cdot)$ to normalize the channel dimension of $\boldsymbol{Q}_t$, which, similar to [37], can stabilize the value of memory cells and help to prevent the vanishing/exploding gradients (see Appendix C for details).

Figure 2: Illustration of generating the memory cell convolution kernel, where (a) is for 2D tensors and (b) for 3D tensors.

The idea of dynamically generating network weights has been used in many works [6, 14, 15, 23, 44, 46], where in [14] location-dependent convolutional kernels are also dynamically generated to improve CNNs. In contrast to these works, we focus on broadening the receptive field of $t$LSTM memory cells. Whilst the flexibility is retained, fewer parameters are required to generate the kernel since the kernel is shared by different memory cell channels.

**Channel Normalization.** To improve training, we adapt Layer Normalization (LN) [3] to our $t$LSTM. Similar to the observation in [3] that LN does not work well in CNNs where channel vectors at different locations have very different statistics, we find that LN is also unsuitable for $t$LSTM where lower level information is near the input while higher level information is near the output. We

therefore normalize the channel vectors at different locations with their own statistics, forming a *Channel Normalization (CN)*, with its operator CN $(\cdot)$:

$$CN\,(\boldsymbol{Z}; \boldsymbol{\Gamma}, \boldsymbol{B}) = \widehat{\boldsymbol{Z}} \odot \boldsymbol{\Gamma} + \boldsymbol{B} \tag{20}$$

where $\boldsymbol{Z}, \widehat{\boldsymbol{Z}}, \boldsymbol{\Gamma}, \boldsymbol{B} \in \mathbb{R}^{P \times M^z}$ are the original tensor, normalized tensor, *gain* parameter, and *bias* parameter, respectively. The $m^z$-th channel of $\boldsymbol{Z}$, i.e. $\boldsymbol{z}_{m^z} \in \mathbb{R}^P$, is normalized element-wisely:

$$\widehat{\boldsymbol{z}}_{m^z} = (\boldsymbol{z}_{m^z} - \boldsymbol{z}^\mu)/\boldsymbol{z}^\sigma \tag{21}$$

where $\boldsymbol{z}^\mu, \boldsymbol{z}^\sigma \in \mathbb{R}^P$ are the *mean* and *standard deviation* along the channel dimension of $\boldsymbol{Z}$, respectively, and $\widehat{\boldsymbol{z}}_{m^z} \in \mathbb{R}^P$ is the $m^z$-th channel of $\widehat{\boldsymbol{Z}}$. Note that the number of parameters introduced by CN/LN can be neglected as it is very small compared to the number of other parameters in the model.

**Using Higher-Dimensional Tensors.** One can observe from (9) that when fixing the kernel size $K$, the tensor size $P$ of a 2D *t*LSTM grows linearly w.r.t. its depth $L$. How can we expand the tensor volume more rapidly so that the network can be widened more efficiently? We can achieve this goal by leveraging higher-dimensional tensors. Based on previous definitions for 2D *t*LSTMs, we replace the 2D tensors with $D$-dimensional ($D > 2$) tensors, obtaining $\boldsymbol{H}_t, \boldsymbol{C}_t \in \mathbb{R}^{P_1 \times P_2 \times \ldots \times P_{D-1} \times M}$ with the tensor size $\boldsymbol{P} = [P_1, P_2, \ldots, P_{D-1}]$. Since the hidden states are no longer matrices, we concatenate the projection of $\boldsymbol{x}_t$ to one *corner* of $\boldsymbol{H}_{t-1}$, and thus (5) is extended as:

$$\boldsymbol{h}_{t-1,\boldsymbol{p}}^{cat} = \begin{cases} \boldsymbol{x}_t \boldsymbol{W}^x + \boldsymbol{b}^x & \text{if } p_d = 1 \text{ for } d = 1, 2, \ldots, D-1 \\ \boldsymbol{h}_{t-1,\boldsymbol{p-1}} & \text{if } p_d > 1 \text{ for } d = 1, 2, \ldots, D-1 \\ \boldsymbol{0} & \text{otherwise} \end{cases} \tag{22}$$

where $\boldsymbol{h}_{t-1,\boldsymbol{p}}^{cat} \in \mathbb{R}^M$ is the channel vector at location $\boldsymbol{p} \in \mathbb{Z}_+^{D-1}$ of the concatenated hidden state $\boldsymbol{H}_{t-1}^{cat} \in \mathbb{R}^{(P_1+1) \times (P_2+1) \times \ldots \times (P_{D-1}+1) \times M}$. For the tensor update, the convolution kernel $\boldsymbol{W}^h$ and $\boldsymbol{W}_t^c(\cdot)$ also increase their dimensionality with kernel size $\boldsymbol{K} = [K_1, K_2, \ldots, K_{D-1}]$. Note that $\boldsymbol{W}_t^c(\cdot)$ is reshaped from the vector, as illustrated in Fig. 2(b). Correspondingly, we generate the output $\boldsymbol{y}_t$ from the opposite *corner* of $\boldsymbol{H}_{t+L-1}$, and therefore (8) is modified as:

$$\boldsymbol{y}_t = \varphi(\boldsymbol{h}_{t+L-1,\boldsymbol{P}} \boldsymbol{W}^y + \boldsymbol{b}^y) \tag{23}$$

For convenience, we set $P_d = P$ and $K_d = K$ for $d = 1, 2, \ldots, D-1$ so that all dimensions of $\boldsymbol{P}$ and $\boldsymbol{K}$ can satisfy (9) with the same depth $L$. In addition, CN still normalizes the channel dimension of tensors.

## 3 Experiments

We evaluate *t*LSTM on five challenging sequence learning tasks under different configurations:

(a) *s***LSTM (baseline)**: our implementation of *s*LSTM [21] with parameters shared across all layers.

(b) **2D *t*LSTM**: the standard 2D *t*LSTM, as defined in (14)-(19).

(c) **2D *t*LSTM–M**: removing (–) memory (M) cell convolutions from (b), as defined in (10)-(13).

(d) **2D *t*LSTM–F**: removing (–) feedback (F) connections from (b).

(e) **3D *t*LSTM**: tensorizing (b) into 3D *t*LSTM.

(f) **3D *t*LSTM+LN**: applying (+) LN [3] to (e).

(g) **3D *t*LSTM+CN**: applying (+) CN to (e), as defined in (20).

To compare different configurations, we also use $L$ to denote the number of layers of a *s*LSTM, and $M$ to denote the hidden size of each *s*LSTM layer. We set the kernel size $K$ to 2 for 2D *t*LSTM–F and 3 for other *t*LSTMs, in which case we have $L = P$ according to (9).

For each configuration, we fix the parameter number and increase the tensor size to see if the performance of *t*LSTM can be boosted without increasing the parameter number. We also investigate how the runtime is affected by the depth, where the runtime is measured by the average GPU milliseconds spent by *a forward and backward pass over one timestep of a single example*. Next, we compare *t*LSTM against the state-of-the-art methods to evaluate its ability. Finally, we visualize the internal working mechanism of *t*LSTM. Please see Appendix D for training details.

## 3.1  Wikipedia Language Modeling

The Hutter Prize Wikipedia dataset [25] consists of 100 million characters taken from 205 different characters including alphabets, XML markups and special symbols. We model the dataset at the character-level, and try to predict the next character of the input sequence.

We fix the parameter number to 10M, corresponding to channel sizes $M$ of 1120 for $s$LSTM and 2D $t$LSTM–F, 901 for other 2D $t$LSTMs, and 522 for 3D $t$LSTMs. All configurations are evaluated with depths $L = 1, 2, 3, 4$. We use Bits-per-character (BPC) to measure the model performance.

Results are shown in Fig. 3. When $L \leq 2$, $s$LSTM and 2D $t$LSTM–F outperform other models because of a larger $M$. With $L$ increasing, the performances of $s$LSTM and 2D $t$LSTM–M improve but become saturated when $L \geq 3$, while $t$LSTMs with memory cell convolutions improve with increasing $L$ and finally outperform both $s$LSTM and 2D $t$LSTM–M. When $L = 4$, 2D $t$LSTM–F is surpassed by 2D $t$LSTM, which is in turn surpassed by 3D $t$LSTM. The performance of 3D $t$LSTM+LN benefits from LN only when $L \leq 2$. However, 3D $t$LSTM+CN consistently improves 3D $t$LSTM with different $L$.

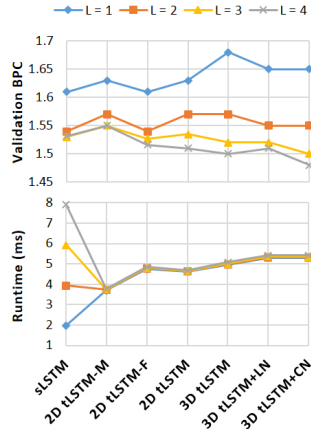

Figure 3: Performance and runtime of different configurations on Wikipedia.

Whilst the runtime of $s$LSTM is almost proportional to $L$, it is nearly constant in each $t$LSTM configuration and largely independent of $L$.

We compare a larger model, i.e. a 3D $t$LSTM+CN with $L = 6$ and $M = 1200$, to the state-of-the-art methods on the test set, as reported in Table 1. Our model achieves 1.264 BPC with 50.1M parameters, and is competitive to the best performing methods [38, 54] with similar parameter numbers.

Table 1: Test BPC on Wikipedia.

|  | BPC | # Param. |
|---|---|---|
| MI-LSTM [51] | 1.44 | ≈17M |
| mLSTM [32] | 1.42 | ≈20M |
| HyperLSTM+LN [23] | 1.34 | 26.5M |
| HM-LSTM+LN [11] | 1.32 | ≈35M |
| Large RHN [54] | 1.27 | ≈46M |
| Large FS-LSTM-4 [38] | 1.245 | ≈47M |
| $2 \times$ Large FS-LSTM-4 [38] | **1.198** | ≈94M |
| 3D $t$LSTM+CN ($L = 6$, $M = 1200$) | 1.264 | 50.1M |

## 3.2  Algorithmic Tasks

(a) **Addition**: The task is to sum two 15-digit integers. The network first reads two integers with one digit per timestep, and then predicts the summation. We follow the processing of [30], where a symbol '-' is used to delimit the integers as well as pad the input/target sequence. A 3-digit integer addition task is of the form:

```
Input:  - 1 2 3 - 9 0 0 - - - - -
Target: - - - - - - - - 1 0 2 3 -
```

(b) **Memorization**: The goal of this task is to memorize a sequence of 20 random symbols. Similar to the addition task, we use 65 different

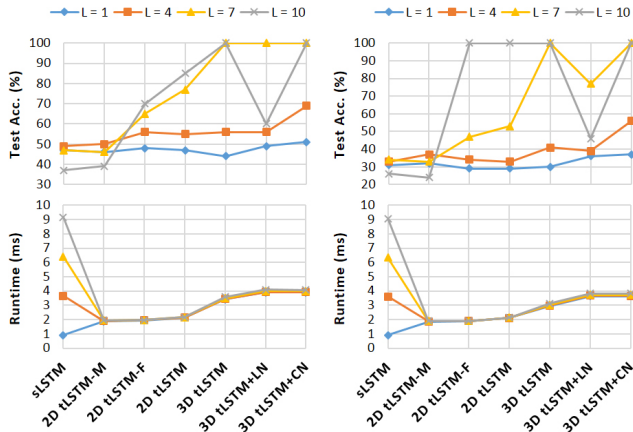

Figure 4: Performance and runtime of different configurations on the addition (left) and memorization (right) tasks.

symbols. A 5-symbol memorization task is of the form:

```
Input:   - a b c c b - - - - - -
Target:  - - - - - - a b c c b -
```

We evaluate all configurations with $L = 1, 4, 7, 10$ on both tasks, where $M$ is 400 for *addition* and 100 for *memorization*. The performance is measured by the symbol prediction accuracy.

Fig. 4 show the results. In both tasks, large $L$ degrades the performances of *s*LSTM and 2D *t*LSTM–M. In contrast, the performance of 2D *t*LSTM–F steadily improves with $L$ increasing, and is further enhanced by using feedback connections, higher-dimensional tensors, and CN, while LN helps only when $L = 1$. Note that in both tasks, the correct solution can be found (when $100\%$ test accuracy is achieved) due to the repetitive nature of the task. In our experiment, we also observe that for the addition task, 3D *t*LSTM+CN with $L = 7$ outperforms other configurations and finds the solution with only 298K training samples, while for the memorization task, 3D *t*LSTM+CN with $L = 10$ beats others configurations and achieves perfect memorization after seeing 54K training samples. Also, unlike in *s*LSTM, the runtime of all *t*LSTMs is largely unaffected by $L$.

We further compare the best performing configurations to the state-of-the-art methods for both tasks (see Table 2). Our models solve both tasks significantly faster (i.e., using fewer training samples) than other models, achieving the new state-of-the-art results.

Table 2: Test accuracies on two algorithmic tasks.

| | Addition | | Memorization | |
|---|---|---|---|---|
| | Acc. | # Samp. | Acc. | # Samp. |
| Stacked LSTM [21] | 51% | 5M | >50% | 900K |
| Grid LSTM [30] | >99% | 550K | >99% | 150K |
| 3D *t*LSTM+CN ($L=7$) | >99% | **298K** | >99% | 115K |
| 3D *t*LSTM+CN ($L=10$) | >99% | 317K | >99% | **54K** |

### 3.3 MNIST Image Classification

The MNIST dataset [35] consists of 50000/10000/10000 handwritten digit images of size $28 \times 28$ for training/validation/test. We have two tasks on this dataset:

(a) **Sequential MNIST**: The goal is to classify the digit after sequentially reading the pixels in a scan-line order [33]. It is therefore a 784 timestep sequence learning task where a single output is produced at the last timestep; the task requires very long range dependencies in the sequence.

(b) **Sequential Permuted MNIST**: We permute the original image pixels in a fixed random order as in [2], resulting in a permuted MNIST

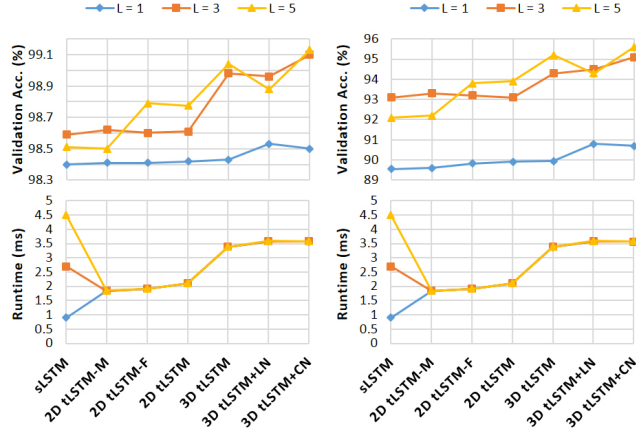

Figure 5: Performance and runtime of different configurations on sequential MNIST (left) and sequential *p*MNIST (right).

(*p*MNIST) problem that has even longer range dependencies across pixels and is harder.

In both tasks, all configurations are evaluated with $M = 100$ and $L = 1, 3, 5$. The model performance is measured by the classification accuracy.

Results are shown in Fig. 5. *s*LSTM and 2D *t*LSTM–M no longer benefit from the increased depth when $L = 5$. Both increasing the depth and tensorization boost the performance of 2D *t*LSTM. However, removing feedback connections from 2D *t*LSTM seems not to affect the performance. On the other hand, CN enhances the 3D *t*LSTM and when $L \geq 3$ it outperforms LN. 3D *t*LSTM+CN with $L = 5$ achieves the highest performances in both tasks, with a validation accuracy of 99.1% for MNIST and 95.6% for *p*MNIST. The runtime of *t*LSTMs is negligibly affected by $L$, and all *t*LSTMs become faster than *s*LSTM when $L = 5$.

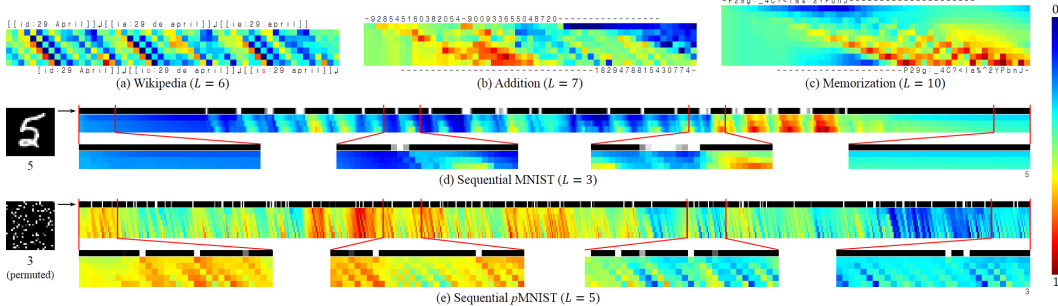

(a) Wikipedia ($L = 6$)  (b) Addition ($L = 7$)  (c) Memorization ($L = 10$)

(d) Sequential MNIST ($L = 3$)

(e) Sequential $p$MNIST ($L = 5$)

Figure 6: Visualization of the diagonal channel means of the $t$LSTM memory cells for each task. In each horizontal bar, the rows from top to bottom correspond to the diagonal locations from $\boldsymbol{p}^{in}$ to $\boldsymbol{p}^{out}$, the columns from left to right correspond to different timesteps (from 1 to $T+L-1$ for the full sequence, where $L-1$ is the time delay), and the values are normalized to be in range $[0, 1]$ for better visualization. Both full sequences in (d) and (e) are zoomed out horizontally.

We also compare the configurations of the highest test accuracies to the state-of-the-art methods (see Table 3). For sequential MNIST, our 3D $t$LSTM+CN with $L=3$ performs as well as the state-of-the-art Dilated GRU model [8], with a test accuracy of 99.2%. For the sequential $p$MNIST, our 3D $t$LSTM+CN with $L=5$ has a test accuracy of 95.7%, which is close to the state-of-the-art of 96.7% produced by the Dilated CNN [40] in [8].

Table 3: Test accuracies (%) on sequential MNIST/$p$MNIST.

|  | MNIST | $p$MNIST |
|---|---|---|
| $i$RNN [33] | 97.0 | 82.0 |
| LSTM [2] | 98.2 | 88.0 |
| $u$RNN [2] | 95.1 | 91.4 |
| Full-capacity $u$RNN [49] | 96.9 | 94.1 |
| $s$TANH [53] | 98.1 | 94.0 |
| BN-LSTM [13] | 99.0 | 95.4 |
| Dilated GRU [8] | **99.2** | 94.6 |
| Dilated CNN [40] in [8] | 98.3 | **96.7** |
| 3D $t$LSTM+CN ($L=3$) | **99.2** | 94.9 |
| 3D $t$LSTM+CN ($L=5$) | 99.0 | 95.7 |

## 3.4 Analysis

The experimental results of different model configurations on different tasks suggest that the performance of $t$LSTMs can be improved by increasing the tensor size and network depth, requiring no additional parameters and little additional runtime. As the network gets wider and deeper, we found that the memory cell convolution mechanism is crucial to maintain improvement in performance. Also, we found that feedback connections are useful for tasks of sequential output (e.g., our Wikipedia and algorithmic tasks). Moreover, $t$LSTM can be further strengthened via tensorization or CN.

It is also intriguing to examine the internal working mechanism of $t$LSTM. Thus, we visualize the memory cell which gives insight into how information is routed. For each task, the best performing $t$LSTM is run on a random example. We record the channel mean (the mean over channels, e.g., it is of size $P \times P$ for 3D $t$LSTMs) of the memory cell at each timestep, and visualize the diagonal values of the channel mean from location $\boldsymbol{p}^{in}=[1, 1]$ (near the input) to $\boldsymbol{p}^{out}=[P, P]$ (near the output).

Visualization results in Fig. 6 reveal the distinct behaviors of $t$LSTM when dealing with different tasks: (i) Wikipedia: the input can be carried to the output location with less modification if it is sufficient to determine the next character, and vice versa; (ii) addition: the first integer is gradually encoded into memories and then interacts (performs addition) with the second integer, producing the sum; (iii) memorization: the network behaves like a shift register that continues to move the input symbol to the output location at the correct timestep; (iv) sequential MNIST: the network is more sensitive to the pixel value change (representing the contour, or topology of the digit) and can gradually accumulate evidence for the final prediction; (v) sequential $p$MNIST: the network is sensitive to high value pixels (representing the foreground digit), and we conjecture that this is because the permutation destroys the topology of the digit, making each high value pixel potentially important.

From Fig. 6 we can also observe common phenomena for all tasks: (i) at each timestep, the values at different tensor locations are markedly different, implying that wider (larger) tensors can encode more information, with less effort to compress it; (ii) from the input to the output, the values become increasingly distinct and are shifted by time, revealing that deep computations are indeed performed together with temporal computations, with long-range dependencies carried by memory cells.

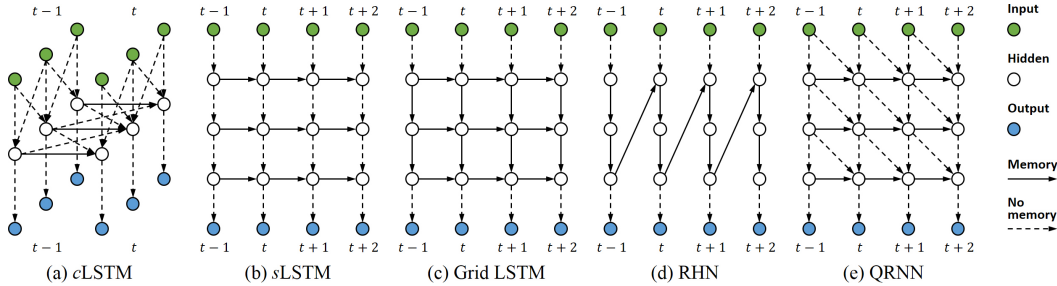

Figure 7: Examples of models related to *t*LSTMs. (a) A single layer *c*LSTM [48] with vector array input. (b) A 3-layer *s*LSTM [21]. (c) A 3-layer Grid LSTM [30]. (d) A 3-layer RHN [54]. (e) A 3-layer QRNN [7] with kernel size 2, where costly computations are done by temporal convolution.

## 4 Related Work

**Convolutional LSTMs.** Convolutional LSTMs (*c*LSTMs) are proposed to parallelize the computation of LSTMs when the input at each timestep is *structured* (see Fig. 7(a)), e.g., a vector array [48], a vector matrix [41, 42, 50, 52], or a vector tensor [9, 45]. Unlike *c*LSTMs, *t*LSTM aims to increase the capacity of LSTMs when the input at each timestep is *non-structured*, i.e., a single vector, and is advantageous over *c*LSTMs in that: (i) it performs the convolution across different hidden layers whose structure is independent of the input structure, and integrates information bottom-up and top-down; while *c*LSTM performs the convolution within each hidden layer whose structure is coupled with the input structure, thus will fall back to the vanilla LSTM if the input at each timestep is a single vector; (ii) it can be widened efficiently without additional parameters by increasing the tensor size; while *c*LSTM can be widened by increasing the kernel size or kernel channel, which significantly increases the number of parameters; (iii) it can be deepened with little additional runtime by delaying the output; while *c*LSTM can be deepened by using more hidden layers, which significantly increases the runtime; (iv) it captures long-range dependencies from multiple directions through the memory cell convolution; while *c*LSTM struggles to capture long-range dependencies from multiple directions since memory cells are only gated along one direction.

**Deep LSTMs.** Deep LSTMs (*d*LSTMs) extend *s*LSTMs by making them deeper (see Fig. 7(b)-(d)). To keep the parameter number small and ease training, Graves [22], Kalchbrenner *et al.* [30], Mujika *et al.* [38], Zilly *et al.* [54] apply another RNN/LSTM along the *depth* direction of *d*LSTMs, which, however, multiplies the runtime. Though there are implementations to accelerate the deep computation [1, 16], they generally aim at simple architectures such *s*LSTMs. Compared with *d*LSTMs, *t*LSTM performs the deep computation with little additional runtime, and employs a cross-layer convolution to enable the feedback mechanism. Moreover, the capacity of *t*LSTM can be increased more efficiently by using higher-dimensional tensors, whereas in *d*LSTM all hidden layers as a whole only equal to a 2D tensor (i.e., a stack of hidden vectors), the dimensionality of which is fixed.

**Other Parallelization Methods.** Some methods [7, 8, 28, 29, 36, 40] parallelize the temporal computation of the sequence (e.g., use the temporal convolution, as in Fig. 7(e)) during training, in which case full input/target sequences are accessible. However, during the online inference when the input presents sequentially, temporal computations can no longer be parallelized and will be blocked by deep computations of each timestep, making these methods potentially unsuitable for real-time applications that demand a high sampling/output frequency. Unlike these methods, *t*LSTM can speed up not only training but also online inference for many tasks since it performs the deep computation by the temporal computation, which is also human-like: we convert each signal to an action and *meanwhile* receive new signals in a non-blocking way. Note that for the online inference of tasks that use the previous output $y_{t-1}$ for the current input $x_t$ (e.g., autoregressive sequence generation), *t*LSTM cannot parallel the deep computation since it requires to delay $L-1$ timesteps to get $y_{t-1}$.

## 5 Conclusion

We introduced the Tensorized LSTM, which employs tensors to share parameters and utilizes the temporal computation to perform the deep computation for sequential tasks. We validated our model on a variety of tasks, showing its potential over other popular approaches.

## Acknowledgements

This work is supported by the NSFC grant 91220301, the Alan Turing Institute under the EPSRC grant EP/N510129/1, and the China Scholarship Council.

## Footnotes

*Corresponding authors: Shaobing Gao <gaoshaobing@scu.edu.cn> and Zhen He <hezhen.cs@gmail.com>.

[1]Vectors are assumed to be in row form throughout this paper.

[2]The operator $\langle \cdot \rangle$ returns the cumulative product of all elements in the input variable.

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
