[Supplementary Material]

# A  Mathematical Definition for Cross-Layer Convolutions

## A.1  Hidden State Convolution

The hidden state convolution in (6) is defined as:

$$A_{t,p,m^o} = \sum_{k=1}^{K} \left( \sum_{m^i=1}^{M^i} H^{cat}_{t-1,p-\frac{K-1}{2}+k,m^i} \cdot W^h_{k,m^i,m^o} \right) + b^h_{m^o} \tag{24}$$

where $m^o \in \{1, 2, \cdots, M^o\}$ and zero padding is applied to keep the tensor size.

## A.2  Memory Cell Convolution

The memory cell convolution in (17) is defined as:

$$C^{conv}_{t-1,p,m} = \sum_{k=1}^{K} C_{t-1,p-\frac{K-1}{2}+k,m} \cdot W^c_{t,k,1,1}(p) \tag{25}$$

To prevent the stored information from being flushed away, $C_{t-1}$ is padded with the replication of its boundary values instead of zeros or input projections.

# B  Derivation for the Constraint of $L$, $P$, and $K$

Figure 8: Illustration of calculating the constraint of $L$, $P$, and $K$. Each column is a concatenated hidden state tensor with tensor size $P+1=4$ and channel size $M$. The volume of the output receptive field (blue region) is determined by the kernel radius $K^r$. The output $y_t$ for current timestep $t$ is delayed by $L-1=2$ timesteps.

Here we derive the constraint of $L$, $P$, and $K$ that is defined in (9). The kernel center location is ceiled in case that the kernel size $K$ is not odd. Then, the kernel radius $K^r$ can be calculated by:

$$K^r = \frac{K - K \bmod 2}{2} \tag{26}$$

As shown in Fig. 8, to guarantee the receptive field of $y_t$ covers $x_{1:t}$ while does not cover $x_{t+1:T}$, the following constraint should be satisfied:

$$\tan \angle \text{AOD} \leqslant \tan \angle \text{BOD} < \tan \angle \text{COD} \tag{27}$$

which means:

$$\frac{P}{L} \leqslant \frac{K^r}{1} < \frac{P}{L-1} \tag{28}$$

Plugging (26) into (28), we get:

$$L = \left\lceil \frac{2P}{K - K \bmod 2} \right\rceil \tag{29}$$

# C Memory Cell Convolution Helps to Prevent the Vanishing/Exploding Gradients

Leifert *et al.* [37] have proved that the *lambda gate*, which is very similar to our memory cell convolution kernel, can help to prevent the vanishing/exploding gradients (see Theorem 17-18 in [37]). The differences between our approach and their *lambda gate* are: (i) we normalize the kernel values though a softmax function, while they normalize the gate values by dividing them with their sum, and (ii) we share the kernel for all channels, while they do not. However, as neither modifications affects the conditions of validity for Theorem 17-18 in [37], our memory cell convolution can also help to prevent the vanishing/exploding gradients.

# D Training Details

## D.1 Objective Function

The training objective is to minimize the negative log-likelihood (NLL) of the training sequences w.r.t. the parameter $\boldsymbol{\theta}$ (vectorized), i.e.,

$$\min_{\boldsymbol{\theta}} \frac{1}{N} \sum_{n=1}^{N} \sum_{t=1}^{T_n} -\ln p(\boldsymbol{y}_{n,t}^d | f(\boldsymbol{x}_{n,1:t}^d; \boldsymbol{\theta})) \tag{30}$$

where $N$ is the number of training sequences, $T_n$ the length of the $n$-th training sequence, and $p(\boldsymbol{y}_{n,t}^d | f(\boldsymbol{x}_{n,1:t}^d; \boldsymbol{\theta}))$ the likelihood of target $\boldsymbol{y}_{n,t}^d$ conditioned on its prediction $\boldsymbol{y}_{n,t} = f(\boldsymbol{x}_{n,1:t}^d; \boldsymbol{\theta})$. Since all experiment are classification problems, $\boldsymbol{y}_{n,t}^d$ is represented as the one-hot encoding of the class label, and the output function $\varphi(\cdot)$ is defined as a softmax function, which is used to generate the class distribution $\boldsymbol{y}_{n,t}$. Then, the likelihood can be calculated by $p(\boldsymbol{y}_{n,t}^d | \boldsymbol{y}_{n,t}) = y_{n,t,s}|_{y_{n,t,s}^d=1}$.

## D.2 Common Settings

In all tasks, the NLL (see (30)) is used as the training objective and is minimized by Adam [31] with a learning rate of 0.001. Forget gate biases are set to 4 for image classification tasks and 1 [27] for others. All models are implemented by Torch7 [12] and accelerated by cuDNN on Tesla K80 GPUs.

We only apply CN to the output of the *t*LSTM hidden state as we have tried different combinations and found this is the most robust way that can always improve the performance for all tasks. With CN, the output of hidden state becomes:

$$\boldsymbol{H}_t = \phi\left(\text{CN}\left(\boldsymbol{C}_t; \boldsymbol{\Gamma}, \boldsymbol{B}\right)\right) \odot \boldsymbol{O} \tag{31}$$

## D.3 Wikipedia Language Modeling

As in [10], we split the dataset into 90M/5M/5M for training/validation/test. In each iteration, we feed the model with a mini-batch of 100 subsequences of length 50. During the forward pass, the hidden values at the last timestep are preserved to initialize the next iteration. We terminate training after 50 epochs.

## D.4 Algorithmic Tasks

Following [30], for both tasks we randomly generate 5M samples for training and 100 samples for test, and set the mini-batch size to 15. Training proceeds for at most 1 epoch[3] and will be terminated if $100\%$ test accuracy is achieved.

## D.5 MNIST Image Classification

We set the mini-batch size to 50 and use early stopping for training. The training loss is calculated at the last timestep.

## Footnotes

[3] To simulate the online learning process, we use all training samples only once.