[Reviews · NeurIPS 2017]

Reviewer 1



The paper explores a new way to share parameters in an RNN by using convolution inside an LSTM cell for an unstructured input sequence, and using tensors as convolution kernels and feature maps. The method also adds depth to the model by delaying the output target for a specified number of steps. The idea is quite interesting, and is novel as far as I can tell. The authors provide clear formulation (although a rather complicated one) and provide some experimental results. On a real world dataset (wikipedia LM) the method seems very close to SOA, with about half the parameters. The problems I see with this approach are: - I find it hard to believe that meaningful high dimensional feature maps can be created for most problems, thus scaling to high dimensional tensors is questionable (The authors only try up to dimension 3) - Using “depth in time” introduces a delay and is not suitable for streaming applications (e.g. speech) - For high dimensional tensors the number of hyper parameters can become quite large. Minor nit: - Line 242, it says “Fig.3” and should be “Table 3”

Reviewer 2



The paper introduces a tensorized version of LSTM that allows for implicitly adding depth and width to the network while controlling the computational runtime. The paper is clearly written, the contribution is interesting and the experimental validation is convincing.

Reviewer 3



This paper proposes Tensorized LSTMs for efficient sequence learning. It represents hidden layers as tensors, and employs cross-layer memory cell convolution for efficiency and effectiveness. The model is clearly formulated. Experimental results show the utility of the proposed method. Although the paper is well written, I still have some questions/confusion as follows. I would re-consider my final decision if the authors address these points in rebuttal. 1. My biggest confusion comes from Sec 2.1, when the authors describe how to widen the network with convolution (lines 65-73). As mentioned in text, "P is akin to the number of stacked hidden layers", and the model "locally-connects" along the P direction to share parameters. I think it is a strategy to deepen the network instead of widening it, since increasing P (the number of hidden layers) won't incur more parameters in the convolution. Similarly, as mentioned in lines 103-104, tRNN can be "widened without additional parameters by increasing the tensor size P". It does not make sense, as increasing P is conceptually equivalent to increasing the number of hidden layers in sRNN. This is to deepen the network, not to widen it. 2. The authors claim to deepen the network with delayed outputs (Sec 2.2). They use the parameter L to set up the network depth. However, as shown in Eq. 9, L is determined by P and K, meaning that we can not really set up the network depth as a free parameter. I guess in practice, we would always pre-set P and K before experiments, and then derive L from Eq. 9. It seems over-claimed in lines 6-10, which reads like "we could freely set up the width and depth of the network". 3. The authors claims that the proposed memory cell convolution is able to prevent gradient vanishing/exploding (line 36). This is not verified theoretically or empirically. The words "gradient vanishing/exploding" are even not mentioned in the later text. 4. In the experiments, the authors compared tLSTM variants in the following dimentions: tensor shape (2D or 3D), normalization (no normalization, LN, CN), memory cell convolution (yes or no), and feedback connections (yes or no). There are 2x3x2x2=24 combinations in total. Why just pick up the six combinations in lines 166-171? I understand it become messy when comparing too many methods, but there are some interesting variants like 2D tLSTM+CN. Also, it might help to split the experiments in groups, like one for normalization strategy, one for memory cell convolution, one for feedback connections, etc.